# DenRAM: neuromorphic dendritic architecture with RRAM for efficient temporal processing with delays

Simone D'Agostino [1,2,3], Filippo Moro[1,2,3], Tristan Torchet[1,3], Yiğit Demirağ[1], Laurent Grenouillet[2], Niccolò Castellani[2], Giacomo Indiveri [1], Elisa Vianello [2] & Melika Payvand [1] ✉

Neuroscience findings emphasize the role of dendritic branching in neocortical pyramidal neurons for non-linear computations and signal processing. Dendritic branches facilitate temporal feature detection via synaptic delays that enable coincidence detection (CD) mechanisms. Spiking neural networks highlight the significance of delays for spatio-temporal pattern recognition in feed-forward networks, eliminating the need for recurrent structures. Here, we introduce DenRAM, a novel analog electronic feed-forward spiking neural network with dendritic compartments. Utilizing 130 nm technology integrated with resistive RAM (RRAM), DenRAM incorporates both delays and synaptic weights. By configuring RRAMs to emulate bio-realistic delays and exploiting their heterogeneity, DenRAM mimics synaptic delays and efficiently performs CD for pattern recognition. Hardware-aware simulations on temporal benchmarks show DenRAM's robustness against hardware noise, and its higher accuracy over recurrent networks. DenRAM advances temporal processing in neuromorphic computing, optimizes memory usage, and marks progress in low-power, real-time signal processing

The dendritic tree of biological neurons is an intriguing and prominent structure, with multiple branches (arbors) hosting several clusters of synapses, enabling communication and processing in complex networks. Communication among neurons involves the reception of action potentials (spikes) at the synapse level, generation of a post-synaptic current, with amplitude proportional to the synaptic weight, and propagation of the weighted sum of all the currents to the cell body (the soma). Spikes are produced in the neuron soma and transmitted through its axon to the synapses of the destination neurons (Fig. 1a). Dendritic arbors exhibit a wide range of behaviors useful for computation, such as spatio-temporal feature detection, low-pass filtering, and non-linear integration[1]. Therefore, it has been suggested that dendritic branches can be considered as semi-independent computing units in the brain[2–5]. Despite these remarkable features of dendrites, most neuron models used in Artificial Neural Networks

(ANNs) do not take them into account, and instead use the so-called "point neurons" (i.e., neurons with all synapses connected to the same node, with no spatial differentiation). While ANNs with point-neuron models can produce remarkable results for static inputs (or for discrete sequences of static inputs), they are not ideal for processing the temporal aspects of dynamic input patterns. Indeed, it has been recently shown that to replicate the properties of a single cortical neuron with dendritic arborization it is required to use deep ANNs of 5–8 layers[6].

A key feature of dendrites is their ability to detect local features through the spatial and temporal alignment of synapse activations within a branch, known as Coincidence Detection (CD)[7,8]. CD can capture temporal signal features across various time scales, effectively turning dendrites into multi-time scale processing units[9]. The spatial arrangement of synapses on dendrites influences both local and

[1]Institute of Neuroinformatics, University of Zurich and ETH Zurich, Zurich, Switzerland. [2]CEA-Leti, Université Grenoble Alpes, Grenoble, France. [3]These authors contributed equally: Simone D'Agostino, Filippo Moro, Tristan Torchet. ✉e-mail: melika@ini.uzh.ch

somatic responses, with functionally related synapses forming clusters to enhance feature detection, ultimately improving computational efficiency and storage capacity[10]. The spatial arrangement of synapses on dendrites can be modeled using neuron compartments (Fig. 1b), where each compartment acts as a spatio-temporal processing element.

Spatio-temporal feature detection is found in many biological information processing circuits, such as the Barn owl's sound source localization using the Interaural Time Difference[11–13], or the vibration source detection by the sand scorpion[14,15]. These biological systems encode information in the precise timing of spike events, allowing for extremely accurate sensing at low power consumption. Since the time of arrival of the events is at the core of this computation, temporal variables, such as delays play a significant role in computation.

Previous studies in Spiking Neural Networks (SNNs) have demonstrated that training temporal variables, such as a synapse and neuron time constant, can enhance the accuracy of network in classifying spatio-temporal patterns[16–20]. More recently, delays have garnered increased attention as temporal variables that enrich the computational efficiency of SNNs[21–24]. As illustrated in Fig. 1c, state-of-the-art architectures for tasks such as classifying the Spiking Heidelberg Digit (SHD) dataset[25] (a keyword spotting task) achieve the highest accuracy by incorporating delays as additional network parameters. In contrast, recurrent architectures perform poorly and require a larger number of parameters (Fig. 1d). Such sensory processing tasks require delays in the time scales of 10s–100s milliseconds, and sometimes even seconds.

To perform real-time sensory processing applications on the edge using neuromorphic SNN hardware (Fig. 1e), on-chip representation of such delay time scales is required. Examples of neuromorphic chips that have dendritic circuits integrated with silicon neurons have been presented in the past[26,27]. Indeed, previous work has shown how these types of architectures can detect spatio-temporal patterns on-chip[27–31]. However, most of these either do not use delays as a variable for computation, or use circuits at accelerated time-scales that do not support closed-loop real-time processing.

Implementation of delays are costly because each synapse requires an additional memory element and a different set of network parameters. A key to this memory is its short-term dynamics, to keep the information of the incoming spike for the required amount of time.

Previous implementations of on-chip delays using Complementary Metal-Oxide-Semiconductor (CMOS) technology have used digital buffers[23], active analog circuits[28,32,33], or mixed-signal solutions[26]. On the other hand, emerging memory technologies, e.g., RRAMs, are promising candidates for implementing these memory elements efficiently, thanks to their non-volatile, small 3D footprint, and zero-static power properties.

While resistive RAM technologies have extensively been used to implement and store the weight parameters of neural networks[34–43], short-term dynamics[44–46], eligibility traces[47], and network connectivity parameters[48], their use for implementing delay elements for edge applications has not yet been fully explored. Recently, we have leveraged the non-volatility and controllable resistive state of RRAM devices as a way to not only implement weights but also efficiently implement delay lines[49].

In this article, we present DenRAM, an RRAM-based dendritic system that has been implemented on chip. We fabricated a prototype dendritic circuit, integrating distributed weights and delay elements, utilizing a hybrid CMOS-RRAM process. The CMOS part of the circuit is manufactured using a low-power 130 nm process, with the RRAM devices fabricated on top of the CMOS foundry layers. The RRAM technology has been specifically developed to implement both

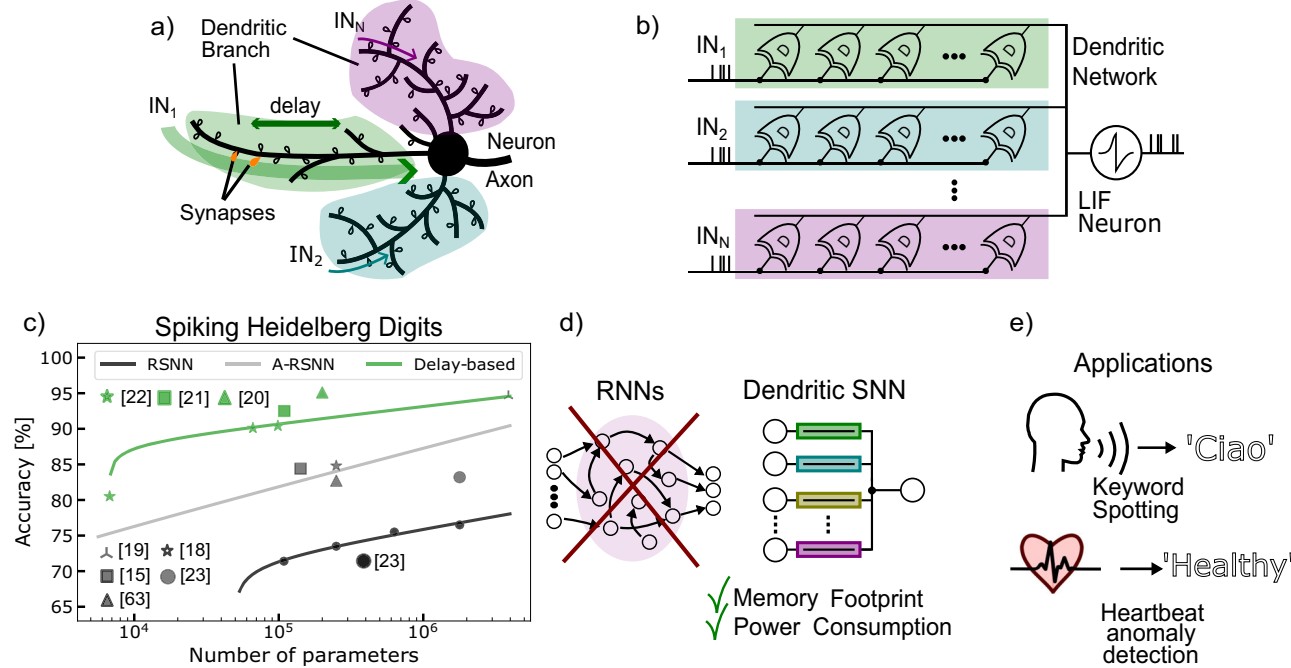

**Fig. 1 | Dendritic RRAM (DenRAM) concept. a** Depiction of a biological Neuron, receiving input spikes through multiple dendritic branches. The packet of neurotransmitters travels across the dendritic branch before reaching the neuron's soma, where it is integrated. **b** Scheme of the Dendritic Network, formed by several Dendritic circuits grouped into Dendritic branches macro-circuits, highlighted by different colors. The branches' outputs are integrated into a Leaky-Integrate-and-Fire Neuron. **c** State-of-the-art results on the SHD dataset as a function of the number of parameters. Delay-based networks show higher accuracy and lower memory footprint compared to recurrent architectures (SRNN: recurrent spiking neural networks, A-SRNN: augmented-SRNN). **d** Recurrent Neural Networks are hard to train and yield low performance. Dendritic SNNs are feed-forward models that perform better than RNNs despite reduced Memory Footprint and Power Consumption. **e** Applications for the Dendritic SNN include Key-Word-Spotting and Heartbeat anomaly detection, and possibly many other sequence processing tasks. Illustrations in **a**, **d**, and **e** were created with Inkscape.

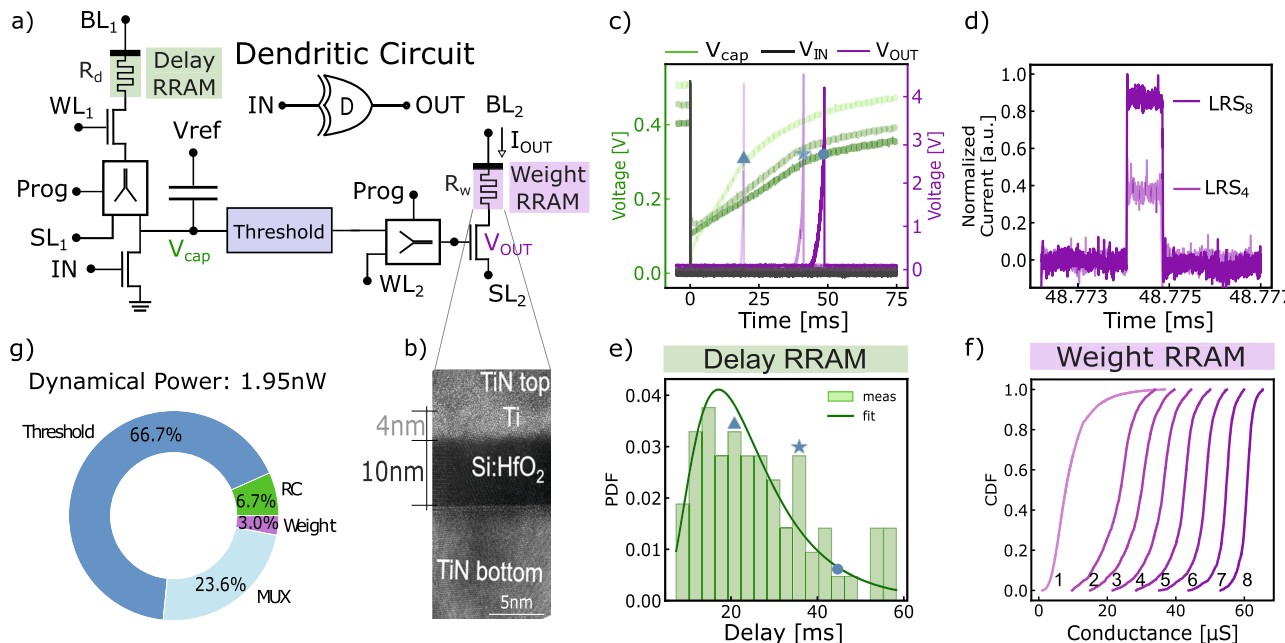

**Fig. 2 | Dendritic circuit, the building block of the DenRAM architecture.**
**a** Detailed schematics of the Dendritic circuit, featuring the Delay and Weight RRAM devices, a Capacitor, dedicated multiplexers (MUX) for switching between programming and reading operations, and a Threshold unit. **b** Scanning Electron-Microscopy image of a $Hf_xO$-based RRAM device used in the Dendritic circuit. **c** Measurement of the Dendritic Circuit, featuring the voltage on the Capacitor ($V_{cap}$), and output ($V_{OUT}$). The input voltage pulse IN is applied at $t = 0$ s and is not shown in the plot. **d** Probability Distribution Function (PDF) of the delay measurements, with a log-normal distribution fitting curve. **e** Effect of the Weight RRAM on the output current $I_{OUT}$ measured from the Dendritic Circuit. Higher values of conductance (conductance $G_8$ larger than $G_4$, referencing the conductance levels in **f**) increase the output current $I_{OUT}$. **f** Cumulative Distribution Function (CDF) of the Weight RRAM conductance values measured in a 16kb array, in different resistive states. **g** Breakdown of the dynamic power consumption of the dendritic circuit, showing the contributions from all the components in part (**a**). The highest power is attributed to the Threshold block responsible for the 66.7% of the total consumption.

weights and delay lines in hardware. Notably, a thick 10 nm silicon-doped Hafnium Oxide (Si:HfO) layer has been employed, resulting in a wide range of resistance values, approximately six orders of magnitude, when the device transitions from its pristine state to the low resistive state.

We experimentally demonstrate the spatio-temporal feature detection capability (i.e., CD) of DenRAM, and complement it with a novel algorithmic framework that performs the classification of sensory signals strictly using feed-forward connections. By explicitly representing temporal variables through the combination of RRAM-capacitor (RC) elements, DenRAM is capable of preserving temporal information without a need for recurrent connections. Calibrated on our experimental results, we perform hardware-aware simulations to benchmark our approach on two representative edge computing tasks, namely keyword spotting and heartbeat anomaly detection, showing how the introduction of passive delays in the network helps with reducing the memory footprint and power consumption, compared to a recurrent architecture (Fig. 1 d, e).

## Results

### Hardware measurements

**The RRAM-based Dendritic Circuit.** The basic building block of the DenRAM architecture is the synaptic block called "Dendritic Circuit", shown in Fig. 2a. It consists of a delaying unit introduced through the ($R_dC$) combination and a *Threshold* unit, as well as a weighting unit through the ($R_w$) value. The delay and the weighting functionality of the synaptic circuit are both enabled by the use of two RRAM devices per synapse. Each RRAM is connected to its own Bit Line (BL), Word Line (WL), and Source Line (SL), which are controlled to either read or program the memory device. RRAM devices can be either used in the reading mode, or the programming mode, through the use of multiplexer (MUX) circuits. For the Delay RRAM, the MUX *Prog*

selection decides whether the SL is connected to the reading path (transistor with the gate connected to *IN*), or to the programming path $SL_1$. The voltage on the $BL_1$ can be adjusted based on program or read processes. For the Weight RRAM, the MUX *Prog* selection decides whether the gate of the transistor is connected to the reading path from the output of the *Theshold* block, or to the $WL_2$. Voltage on the $SL_2$ can be modified based on programming or reading. More details on the design of the Dendritic Circuit are provided in the Methods section.

We have built the Dendritic Circuit in a 130 nm CMOS process, integrated with RRAM devices in the Back End of the Line (Fig. 2b, more details on the Fabrication Process in the Method section). The fabricated devices have been thoroughly tested in their three main states (Supplementary Note 1, Supplementary Fig. S1): as fabricated, RRAM devices are in the pristine state, where no conductive filament in oxide is present between the two electrodes, thus exhibiting the highest resistance. A one-time forming operation is carried out by applying a positive voltage across the device, causing a conductive filament to form, bringing the device to the Low-Resistive State (LRS). Then, the device becomes programmable between LRS and a High-Resistive State (HRS). Concerning the capacitor in the Dendritic Circuit, we utilize a conventional transistor gate oxide capacitance, setting the capacitance C to around 400 fF.

While in the read mode of the RRAM devices, the *Prog* selector is turned off (0 V), connecting $SL_1$ to the $V_{cap}$ terminal, and the *Threshold* unit output to $WL_2$. $BL_1$ is connected to the source voltage $V_{ref}$, and $BL_2$ is connected to the reading voltage (0.4 V in our experiments). An input voltage pulse of 1.2 V height, and $1\mu s$ of width is applied to the *IN* terminal. This causes the *IN* transistor to conduct, depolarizing the voltage on the capacitor $V_{cap}$ (Fig. 2c, green trace). As soon as the *IN* pulse terminates, the voltage of the capacitor recharges through the Delay RRAM, with the time-constant $\tau = R_dC$, set by $R_d$. After some time, $V_{cap}$ crosses the threshold set by the *Threshold* unit, eliciting a spike,

delayed by $R_dC$ compared to $IN$, at the output of the Dendritic Circuit. Details on the design of the *Threshold* unit is provided in Supplementary Fig. S3a). The delayed pulse is applied to the gate of the Weight RRAM's transistor ($V_{OUT}$), allowing a current $I_{OUT}$ to flow through $R_w$.

**Delay characterization.** We have characterized the delay of the $R_dC$ block in 71 circuits with pristine Delay RRAMs, resulting in the distribution shown in Fig. 2d. Following a log-normal distribution, the mean obtained delay is 22 ms, with a standard deviation of 0.5 ms on the underlying normal distribution. With the conductances of our pristine devices, the minimum delay achieved is 8.08 ms and the maximum is 58.26 ms. Such rather long delays are the result of the particularly large resistance in the Pristine state of Delay RRAMs (Fig. 2c). Crucially, these values match the hypothesized delay produced by dendritic arbors[50,51] and fall in the same order of magnitude as the temporal feature of many temporal tasks, including speech recognition[25], biomedical signal processing[52,53], and robotic control[54].

**Weight characterization.** We provide experimental results showing the effectiveness of the Weight RRAM in modulating the output current $I_{OUT}$. As shown in Fig. 2e, we measure the output current from the Dendritic Circuit in two settings. First, we set $R_w$ to low conductance, with a weak SET operation at 1.6 V, yielding a small output current. Afterward, we set the same device to higher conductance, with a strong SET pulse at 2.0 V, resulting in a much larger current. As illustrated in Fig. 2f, we are capable of modulating the resistance of RRAMs in a broad range of values, enabling the modulation of the output current ($I_{OUT}$). In this case, we modulated the SET programming pulse in 8 distinct levels, obtaining 8 different distributions across a 16 kbit RRAM array (refer to Supplementary Fig. S1 for more details).

**Power consumption characterization.** We characterized the dynamical power consumption with thorough Spice circuit-level simulations, analyzing the contribution of the different components of the circuit (Fig. 2g). In an example simulated scenario, the circuit is fed with a single input spike and produces a single output spike with 30 ms delay which is then weighted by the output RRAM device, set at $R_w = 10$ kΩ. The energy produced to perform such computation is 58.5 pJ, yielding a dynamical power consumption of 1.95 nW. Of such amount, 66.7% is consumed in the *Threshold* block, responsible for capturing the temporal evolution of the $R_dC$ circuit, and producing the delayed spike. The $R_dC$ and $R_w$ are together responsible for less than 10% of the power consumption. The remaining amount is attributed to the MUX selectors. A future iteration of the design will address the energy efficiency of the *Threshold* block, as well as removing the necessity for the MUX selector, further improving the efficiency of the circuit.

**The Dendritic architecture.** The DenRAM architecture consists of an array of the Dendritic Circuits of Fig. 2a, connecting to a downstream Leaky Integrate and Fire (LIF) neurons, as shown in Fig. 3a, featuring two of the $N$ possible dendritic branches, each having multiple synapses. Dendritic Branches can contain as many Dendritic Circuits, with the practical limit being the voltage drop as a result of the wire resistance (known as IR drop), and capacitive loading due to long metal lines (shared Source and Bit Lines). Likewise, many dendritic branches can be stacked together in parallel receiving many input channels/signals. This forms the dendritic tree of a single output neuron. Multiple dendritic trees from different output neurons can be grouped forming a layer of DenRAM that links inputs ($IN_{1,2,...,N}$) to outputs ($OUT_{i,j,...,k}$). Word Lines from all the dendritic trees can be shared in DenRAM.

The same input spike train ($IN_i$) is shared across a dendritic branch, which is a collection of $N$ dendritic circuits. The SL of Weight RRAMs is shared in a branch and, when operating, is connected to the ground. Within a branch, the BL of Delay RRAMs are shared, as well as the BL of Weight RRAM. The current from all the weight RRAMs of one branch sum with the current of the other branches, and is fed to a LIF neuron (see Supplementary Fig. S3b for details on the LIF circuit). As soon as the leaky integration value passes the neuron's threshold, it generates an output spike.

Therefore, the LIF neuron receives a pre-processed version of the inputs, with a large spatio-temporal degree of freedom to adapt to a given task.

DenRAM identifies the temporal features of the input by detecting the coinciding spikes in one branch. With each input spike on each $IN_i$ branch passing through $N$ delays, the neuron's weight parameters can be trained to select the combination of right delays, ensuring spike coincidence (Fig. 3b). Precisely, each dendritic branch takes a spike train $x(t) = \sum_k \delta(t - t_k)$ where $t_k$ are the times at which spike occur. Replicates $x(t)N$ times and introduces different delays, $\Delta_i$, sampled from log-normal distribution, followed by weighing with $w_i$ to obtain $S(t) = \sum_{i=1}^{N} w_i \cdot \sum_k \delta(t - \Delta_i - t_k)$. It computationally performs a temporal correlation-like operation involving counting coincidences of spikes between the original and delayed trains, captured in $S(t)$. This architecture extends beyond the mere processing of immediate inputs; it integrates signals over a temporal spectrum, thereby establishing a dynamic form of short memory, uplifting the neuronal responsiveness to certain temporal sequences.

In real-time sensory processing settings, the temporal features in the environment are in the order of 10s to 100s of ms, which necessitates the Coincidence Detection (CD), and consequently, the delays, to be within the same timing range. This requirement enforces the use of our RRAM devices in their HRS. However, precise control of conductances in HRS is typically much more difficult than in LRS, due to its high variability (Fig. 2f)[55]. In-line with approaches that propose to exploit variability and heterogeneity in neuromorphic circuits for achieving robust computation[56], DenRAM takes advantage of such heterogeneity, by providing a population of analog delay elements per input channel, thus enriching the circuit with a delay spectrum that can be tuned by the weight values.

We perform experiments on the fabricated DenRAM (layout view in Fig. 3c), to showcase its temporal feature detection functionality, through CD. The fabricated DenRAM circuit features 3 input channels or branches, 64 dendritic circuits per branch connected to a single output neuron. The task is to detect a temporal correlation of *T*ms between the spikes of two input channels. In the DenRAM scheme, this means that the delay of *T*ms between two input channels would make the output neuron spike, and thus perform CD. Figure 3d shows the experimental measurements. Two inputs are presented to two different Dendritic Branches ($IN_1$, purple and $IN_2$, blue) with a temporal difference of -60 ms (Fig. 3d, upper plot). To perform CD, the Dendritic Network has to delay the input spike $IN_1$ by a value close to 60 ms using $R_d$, and assign a high weight to its $R_W$, so that the output LIF neuron responds with an output spike due to the coincidence of the $IN_2$ with the delayed version of $IN_1$. We selected four Dendritic Circuits in the first branch ($A$ to $D$), where each of them delays the $IN_1$ spike by a given amount. Dendritic Circuit $D$ produces a delay of 58 ms, making the delay $IN_1$ coincident with $IN_2$ from Dendritic Circuit $A$. Therefore, we program the RRAM Weight of the G circuit, $R_{W,G}$ to LRS to maximize the current injected into the output neuron at the coincidence. We set all the other Dendritic Circuits' Weight RRAMs to HRS. This is reflected in the measurements of the currents from the Dendritic Circuits (Fig. 3d, middle plot). The current from the coincident spikes of $D_1$ and $A_2$ are the highest, due to the corresponding $R_w$ at their LRS, and the rest have lower currents. These currents are integrated by the neuron, and its membrane voltage is measured, shown in Fig. 3d, lower plot. It can be seen that the neuron responds maximally to the coincidence of spikes coming from Dendritic Circuit $D_1$ and $IN_2$, correctly performing CD. To ensure the robustness of this functionality, we

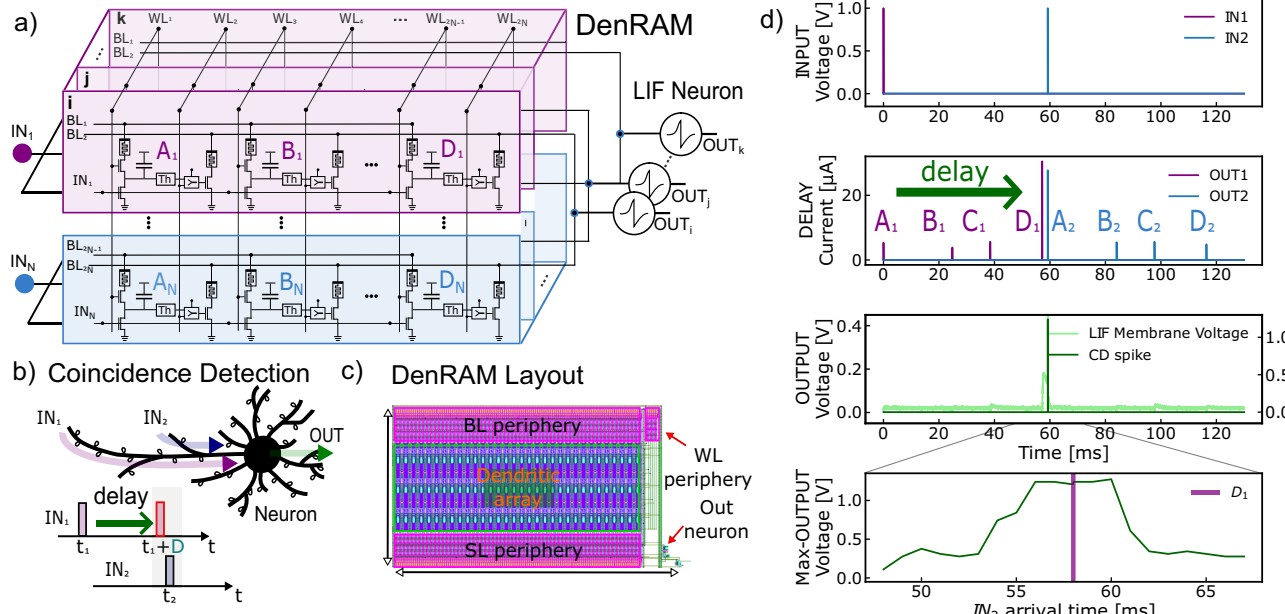

**Fig. 3 | Coincidence Detection with Dendritic Circuits. a** Schematics of the DenRAM architecture. Two input spikes are processed by different dendritic branches, each with different set of delays. The delays that align the two input spikes receive large weights. The inputs are broadcasted to different dendritic trees leading to different output neurons. **b** CD mechanism on a biological neuron, where two inputs are fed from two different synapses and reach the soma at the same time, thanks to dendritic delays. **c** Physical layout of DenRAM. **d** Measurement of the Dendritic Network performing coincidence detection. Illustration in (**b**) created with Inkscape.

varied the temporal difference between the two inputs, such that the network can also separate the two coincident input spikes that are not correlated. The lower plot in Fig. 3d shows the response of the output neuron when the input $IN_2$ changes its arrival time, originally at 60 ms. If $IN_2$ arrives too early or too late compared to $D_1$, the output neuron's voltage will not be fully activated, and coincidence is not detected. An additional measurement changing the weight configuration in Den-RAM is shown in Supplementary Fig. S4.

### Hardware-aware simulations

The biologically inspired temporal delay mechanism implemented with analog resistive RAM brings superior efficiency to DenRAM. In order to test its effectiveness on edge applications with temporally rich inputs, we benchmarked DenRAM on two sets of experiments: heartbeat anomaly detection for biomedical applications, and keyword spotting for audio processing applications. For each benchmark, we evaluated the accuracy, memory footprint, and power consumption of DenRAM, and compared it to a conventional Spiking Recurrent Neural Networks (SRNN) with an equivalent accuracy (heartbeat task), and equivalent number of parameters (keyword spotting task). Both tasks require the effective learning of entangled temporal features, which is done by feed-forward synaptic weights of DenRAM that learn to utilize their corresponding synaptic delays.

To deliver software-comparable accuracy on the benchmarks with our analog substrate, we resorted to several hardware-aware training procedures[40,57] (see Methods for details).

For the weights, to account for the imprecise programming operation of RRAM devices and RRAM conductance relaxation[58] (which restrains the mapping of full-precision simulation weights to RRAM devices), we utilized noise-aware training[40] by injecting a Gaussian noise to trained network weights using the "straight-through estimator" technique. The model of the noise is derived by conductance measurements of our programmed RRAM devices after a relaxation period (Fig. 2f) and set to 10% of the maximum conductance value in the layer[59]. This approach makes sure that the obtained simulation weights can be mappable on devices with minimal accuracy drop after realistic programming stochasticity and device relaxation effects.

For the delays, we used the same log-normal distribution shown in Fig. 2d to sample, and then fixed the delays throughout the simulation for the heartbeat anomaly detection task. However, our experiments with the keyword spotting task below demonstrated that speech signals require longer delays than what could be implemented with our proposed circuit (<60 ms). Therefore for SHD, we use a log-normal distribution with a higher mean (mean of 500 ms for the highest accuracy) to sample and fix delays to solve the task. This approach provides guidance for what could be achieved using higher-resistance RRAM devices in DenRAM neuromorphic architecture.

It is worth noting that in our scheme, delays are not trained and RRAM weight parameters are the only trainable parameters.

Nevertheless, the network will learn to weight randomly delayed versions of the input for performing optimal CD to detect the temporal features.

### Heartbeat anomaly detection task.

We first benchmark DenRAM on a heartbeat anomaly detection task using the Electrocardiography (ECG) recordings of the MIT-BIH Arrhythmia Database[52]. This is a binary classification task to distinguish between a healthy heartbeat and an arrhythmia. For the data to be compatible with DenRAM, we first encode the continuous ECG time-series into trains of spikes using a delta-modulation technique, which describes the relative changes in signal magnitude[60,61] (see Methods for details on the dataset). This encoding scheme produces two trains of spikes for a single streaming input, one corresponding to the positive, and the other to the negative changes of the input. Therefore, the network requires only two dendritic arbors corresponding to the $IN_1$ and $IN_2$ inputs in Fig. 3a.

We train the network with 10% noise injection on the RRAM weights, sweep the number of synapses per channel, and evaluate the accuracy of the network on the test set. The results are plotted in Fig. 4a. DenRAM is plotted using the green curve, which is compared against the SRNN shown in the purple curve. The error bar is variability across 5 seeds.

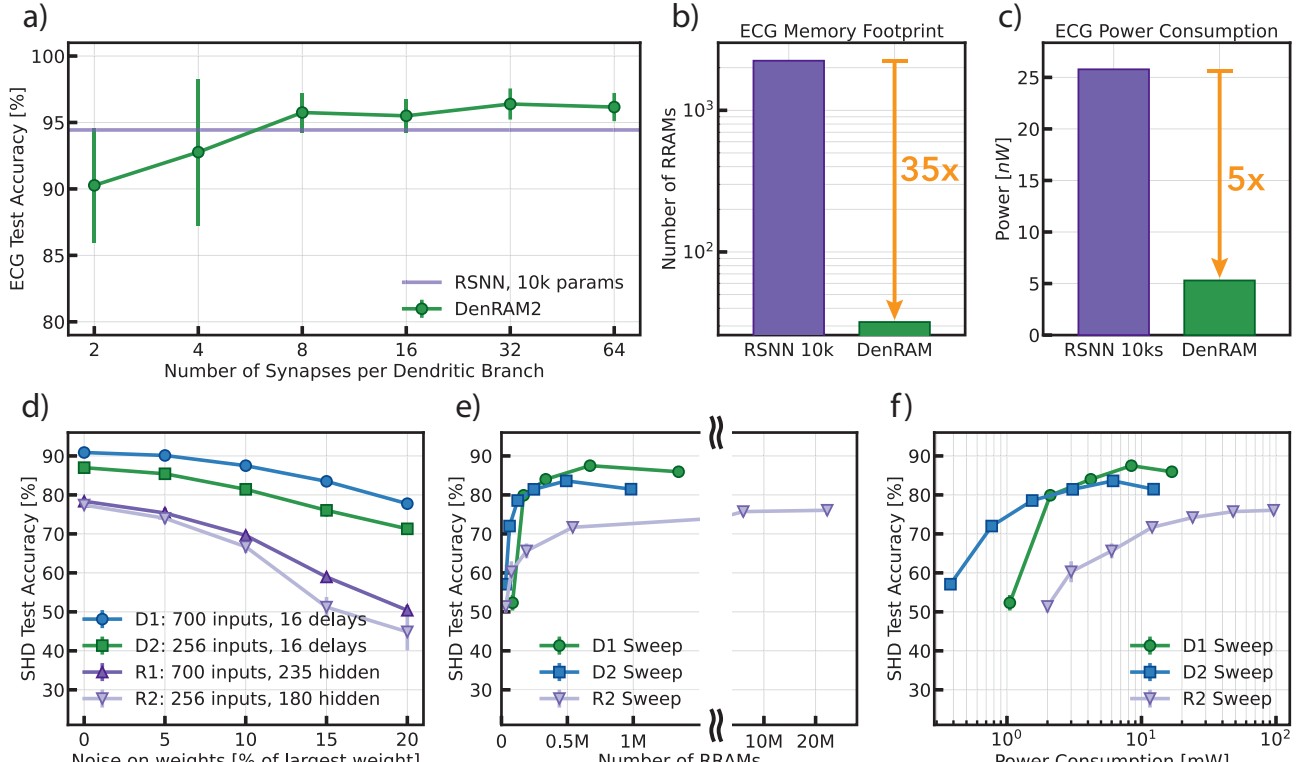

**Fig. 4 | Performance of DenRAM on Heartbeat Anomaly Detection and Keyword Spotting. a** Classification accuracy as a function of the number of synapses per dendritic branch and comparison with a SRNN of 32 neurons (1.1k parameters). Error bars capture the standard deviation over 10 trials. **b** Memory Footprint of the DenRAM architecture solving ECG, compared with an iso-accuracy SRNN. **c** Power consumption of DenRAM in the ECG task and comparison an iso-accuracy SRNN. **d** Classification accuracy as a function of the noise introduced on the weights for two delay architectures (D1: 700 inputs, 16 delays; D2: 256 inputs, 16 delays) and for two SRNN architectures with one hidden layer (R1: 700 inputs, 235 hidden neurons; R2: 256 inputs, 180 hidden neurons). **e** Classification accuracy (with RRAM-calibrated noise on the weights) concerning the network's number of parameters for D1, D2 and R2, sweeping the number of synapses per branch for D1 and D2, the number of hidden neurons for R2. **f** Power consumption of each network configuration (D1, D2 and R2) shown in **e**. In **d–f** error bars represent the standard deviation over 3 trials.

DenRAM is capable of solving this temporal task with a mean accuracy of 95.30%, with 8 synapses (weights and delays) per dendritic branch, corresponding to 16 parameters and 64 devices. The network size is 35 times smaller than the iso-accuracy SRNN made of an input layer of 2 neurons, a recurrent layer of 32 neurons, and 2 fully connected output neurons (a total of 1152 parameters, 2304 devices). It is worth noting that since the network is trained using the delay distribution of our measurements in Fig. 2d, this task is solvable with a mean delay of 22 ms.

Avoiding the explicit recurrence in SRNN allows DenRAM to utilize its parameters more efficiently. While the recurrent parameters in SRNNs are tuned to create complex dynamics of neuron activity, DenRAM leverages delay to explicitly process temporal features. The parameters in the recurrent layer of the SRNN scale quadratically with the number of neurons, while parameters scale linearly with input size in DenRAM. This yields a great advantage for DenRAM in the model's Memory Footprint, as highlighted in Fig. 4b.

DenRAM's efficient temporal processing is also reflected in the estimation of power consumption. To estimate DenRAM's power consumption, we extend the Spice simulation presented in Fig. 2g to the system level, yielding a power consumption of 5.30 nW during inference. We compare DenRAM with an implementation of an iso-accuracy SRNN built with the same 130 nm RRAM-enhanced technological substrate. The iso-accuracy SRNN features 32 hidden neurons and around 1.1k parameters (see Supplementary Fig. 5d). We assume efficient implementation of LIFneurons (see Supplementary Fig. 3b) and run Spice simulations based on this circuit. The

result reveals a reduction of **5** times in power in favor of the DenRAM architecture.

**Keyword spotting.** We next benchmark DenRAM on a more complex task of keyword spotting using the SHD dataset, which consists of spoken digits of 20 classes, fed through 700 Mel-spaced band-pass filters, whose output is encoded into spikes (see Methods). To compare the generalization performance of DenRAM compared to SRNN, we implemented four networks: D1 (DenRAM with 16 delays per channel, using 700 input channels), R1 (a SRNN with 700 input channels and 235 hidden neurons), and their hardware-optimized counterparts, with smaller size: D2 (a network with 16 delays per channel, using 256 input channels), and R2 (a SRNN with 256 input channels and 180 hidden neurons). For the details of the sub-sampling of D2 and R2, refer to the Methods. D1/R1 and D2/R2 have the same number of trainable parameters: 224k in the former case, and 82k in the latter case. Each trainable parameter translates to two RRAM devices to implement the weight (positive and negative weights) in both DenRAM and SRNN, while DenRAM also features an additional Delay RRAM per dendritic circuit, which is not a trainable parameter. We first compare the four networks in their resilience against the analog RRAM noise. Fig. 4d shows that DenRAM architectures of D1 and D2 are consistently better performing compared to the SRNN of R1 and R2, for any amount of noise injected into the network. D1 is the best-performing architecture with 90.88% without noise, and is resilient to noise, with only 3.3% of drop in accuracy for up to 10% of injected noise on the RRAM weights, compatible with our hardware. This is compared to a drop of

8.75% in accuracy, in the case of R1 for the same amount of noise (from 78.37% in the noise-less case). On the smaller networks with reduced memory footprint, D2 and R2 accuracies achieve 81.43% and 66.7%, confirming that the delay-based architectures provide more expressive representations even with less number of parameters.

Next, we evaluate the accuracy of our networks as a function of the number of parameters in the network by sweeping the number of parameters of the D1, D2 and R2 networks, with 10% of noise on the RRAM weights. Figure 4e shows that the DenRAM network has consistently better accuracy compared to the SRNN architecture, for the same number of parameters. D1 accuracy reaches the highest accuracy of 87.5% using 224k parameters (700 input channels and 16 delays per channel) while injecting 10% of RRAM noise. To the best of our knowledge, this is the highest accuracy achieved on the SHD dataset, taking into account the variability of the analog substrate (up to 10%). Comparatively, R1 reaches 69.62% accuracy, with an equivalent number of parameters and noise conditions. We suspect this big performance drop in SRNN might be due to the higher non-linear effects of the noise in the SRNN as compared to the DenRAM, due to the recurrent connections.

Finally, we evaluate the accuracy of D1, D2, and R2 networks by varying their maximum power consumption, as determined from the power estimations presented in Fig. 2g. The results are depicted in Fig. 4f, where it is observed that the accuracy of the R2 network plateaus at approximately 76%, regardless of increasing power consumption. In contrast, the D2 network achieves an accuracy of 83.6% at a power consumption of 6.15 μW, while the D1 network attains an accuracy of 87.5% with a power consumption of 8.41 μW.

## Discussion
### Perspectives on the role of delays in temporal data processing
Delays are at the core of the spatio-temporal processing performed by dendrites and their central importance has been demonstrated in DenRAM. We have shown that delays, and dendritic computation in general, are key to improving computational and energy efficiency of spiking neural network chips. Delays explicitly enable the construction of temporal computational primitives and allow avoiding/reducing recurrent connections to induce temporal dynamic processing[62]. By providing feed-forward networks the ability to carry out spatio-temporal pattern recognition without having to use recurrent connections we obtained several benefits; Firstly, we could minimize the memory footprint and thus improve the computational efficiency; Secondly, these networks escape the vanishing and exploding gradient problems present in Recurrent Neural Networkss (RNNs) trained with the Backpropagation Through Time (BPTT) algorithm. Both of these benefits are highlighted in Fig. 1c, where delay-based neural networks reached superior accuracy compared to recurrent architectures on the SHD task.

Evidently, for the same number of parameters, delay-based models achieve higher accuracy in the SHD task, confirming the greater computational efficiency of this type of architecture. Crucially, delay-based computation is a relatively novel concept and it is likely that the exploitation of the potential of delays has not yet been maximized. In the DenRAM architecture, similar to[23], the delay parameters are not trained and do not need to be optimized. Methods to adapt delays via gradient descent were recently developed[63], but the exploration of the benefits of training delays is a recent trend[21,22]. In particular, Hammouamri et al. link the delays to temporal causal convolutions, thus making use of modern deep learning techniques to train a SNN, and achieve the highest classification accuracy on the SHD task[21].

Exploiting the techniques that have been developed in deep learning to train such networks is potentially very fruitful. For example, we envision that delays can also implement temporal convolutions[64], and temporal causal Graph Neural Networks[65], where past events can be grouped into a graph representation and achieve impressive results in vision tasks.

Furthermore, the hardware realization of the DenRAM architecture exhibits intriguing parallels with the concept of MLP-mixer, which has demonstrated remarkable performance in vision tasks without relying on convolution or attention mechanisms[66]. While the MLP-mixer randomly mixes features across spatial locations and channels in a two-step process, DenRAM introduces a unique form of temporal shuffling by incorporating random delays to temporal inputs within each channel before integration. These innovative architectural approaches, which reorganize incoming data, whether spatially (as in the case of MLP-mixer) or temporally (as in DenRAM), hold the promise of more effectively extracting complex patterns. This suggests a compelling direction for network design, potentially leading to models that are both computationally efficient and highly capable.

Grounding these novel algorithms on an energy-efficient hardware substrate is thus crucial, and hence is in the the scope of this work. Making an innovative use of the emerging memory technologies expands the functionality of the circuit as well as minimizing power consumption.

### Effect of heterogeneity on the performance of DenRAM
Similar to the computational neuroscience studies on the beneficial role of heterogeneity on the performance of networks[67], we have performed an ablation study on the accuracy of DenRAM on our two benchmarks, by varying the variability of the delay and weight parameters.

We first vary the standard deviation of the underlying normal distribution of delay RRAMs, while fixing the number of synapses per dendritic branch, and fixing the noise of the weight RRAMs to 10%. We find that the required mean delay, to solve the ECG task to 95% accuracy, increases when the standard deviation of the underlying normal distribution decreases. This shows the beneficial role of heterogeneity in loosening the hardware requirements: the higher the variability of the underlying distribution, the lower the required resistance and capacitance of the delay circuit of Fig. 2a) (Supplementary Figs. S5, S6 in Supplementary Note 4).

On the other hand, noise in the programming of RRAM weights was controlled through sweeping the noise injection in terms of percent of the maximum weight in the network. The accuracy of the DenRAM was almost unaffected by noise values up to about 20%, showing to be much more tolerant than the SRNN architecture of iso-accuracy when no noise is injected.

Applying the same methodology used in the delay distribution analysis to the SHD task, we observe parallel outcomes (Supplementary Note 4). This consistency underscores that the characteristics we identified are not unique to a specific task but are, in fact, fundamental attributes of the DenRAM architecture. Furthermore, our research uncovers a significant aspect: the existence of an optimal mean delay within this framework.

Also, we analyzed the effect of unbalancing the number of synapses in each dendritic branch, promoting heterogeneity in the "length" (or size) of dendrites. Setting the Weight-RRAM of a Dendritic Circuit to HRS is equivalent to pruning the dendritic connection, reducing the fan-in of the post-synaptic neuron, and potentially implementing biologically inspired connectivity patterns. We analyzed the effect of heterogeneous dendritic branch size on the performance of DenRAM in Supplementary Fig. S7.

### Delay analysis for keyword spotting
Leveraging both spatial parameters (weights) and temporal parameters (delays), DenRAM effectively utilizes a two-dimensional approach for the separation and classification of spatio-temporal inputs. This capability is elucidated in Supplementary Note 5 and Supplementary Fig. S9, where, in order to discern between input

patterns within the SHD dataset, the network strategically employs two distinct strategies. At times, it opts for "spatial segregation", adjusting the weighting of different channels to differentiate between patterns (e.g., when distinguishing between digits "8" and "18"). Alternatively, it introduces "weight dynamics" through the delayed replications of inputs by dynamically altering the aggregated weight values of each channel over time, thereby implementing "temporal segregation" to distinguish between other patterns (e.g., when differentiating between digits "8" and "17").

## Outlook

**Long time scales represented on chip.** Using RRAM technology, we successfully implemented delay elements that achieved periods of up to 60 ms. This proved sufficient temporal memory for addressing the heartbeat anomaly detection task. However, it fell short in meeting the demands of the keyword spotting task, where delays of up to 500 ms were necessary to reach a desirable accuracy. To accommodate these extended time scales using the same approach, novel technologies are required. Recent research has explored volatile resistive memory devices with tunable time scales, albeit with limitations, typically averaging in the range of 10s of ms[45,46,68]. Material engineering which results in technologies with larger time scale in their decay would prove extremely beneficial in implementing short-term dynamics and delays with minimal area on analog chips. On the non-volatile memory front, Ferroelectric Tunnel Junctions (FTJ)s present an opportunity for the delay element in the next generation of DenRAM, as their resistances is much higher than the RRAM devices (e.g., HRS of FTJ can go beyond GΩ Ohms, as compared to 10–100s of MΩ Ohms for RRAM)[69]. This is due to a different - compared with RRAMs - current conduction mechanism of FTJ devices which is based on tunneling.

**Training delays.** Utilizing the FTJ technology in DenRAM would result in enhanced delays and better control over the delay values in the dendritic branches. This opens to the possibility of learning the delays with gradient descent, as performed in[63]. Optimizing the delay values, it is foreseeable that the number of required Dendritic Circuits per dendritic branch would reduce drastically, further lowering the parameters of DenRAM. As a consequence, we envision that DenRAM could further improve its performance and power efficiency by exploiting novel devices, enabling the training of delays.

**Weight precision.** As highlighted in Fig. 4d, the precision of the weight is an important parameter in determining the performance of Den-RAM. RRAM devices offer relatively good control over the conductance levels by modulating the WL voltage during the SET programming operation. Yet, such programming operations in RRAMs yield a distribution of conductance with about 10% of standard deviation, relative to the maximum conductance (Fig. 2f). To improve the precision of weights in DenRAM, advanced read-verify programming operations on RRAMs can be performed[40,70]. Alternatively, different memory technologies, with better control over their conductance levels, can be used. An example technology is FeFET, demonstrating up to 5-bit precise in its conductance levels[71,72].

**Non-linear integration in dendrites.** Biological dendrites feature multiple characteristics and dynamical behavior that have not been accounted for in this work. For example, the non-linearity of dendritic branches[73] might further enhance the computational capacity of neural networks[74,75] and alone solve XOR task[5]. Such investigations are the natural next step for DenRAM.

**On-chip learning.** The plasticity of resistive RAMs has not been investigated for on-chip learning in this work, despite that being a very promising aspect of such hardware substrate. Dendrites have been shown to play an important role in learning of the cortical circuits[76], and

have been previously implemented in CMOS technology for stochastic on-chip online learning[77]. DenRAM provides an ideal architecture for implementing on-chip learning based on these concepts on RRAM based systems, which is a facet we will be exploring in future work.

As elegantly proposed in a recent study by Boahen[9,78], the concept of dendrocentric computing and learning, where inputs are not only spatially weighted but also temporally considered in accordance with their arrival times, offers a compelling avenue to reduce energy consumption in the next generation of Artificial Intelligence (AI) hardware.

DenRAM stands as a pioneering achievement in this direction, being the first hardware implementation of dendrites that harnesses the unique properties of emerging memory technologies, particularly resistive RAM devices. This marks a significant milestone in advancing dendrocentric computing and learning, setting the stage for more efficient and innovative AI hardware solutions.

## Methods
### Design, fabrication of DenRAM circuits

**Dendritic circuit design.** The Dendritic Circuit (Fig. 2a) features two main sections: one devoted to generating the delay and the second to weight the output currents. The circuit takes input spikes in the form of stereotypical voltage pulses, in our design of 1.2 V peak voltage and 1 μs pulse width. The pulses are applied to the IN terminal, opening the nMOS transistor. The charge on the capacitor, resting at $Vref$, is then pulled to ground during the application of the spike, causing the voltage $Vcap$ to plummet to ground. $Vref$ is set to 0.6 V, a value that maximizes the dynamic range of the capacitor while reducing the read-noise of RRAM devices. As the Multiplexers' selection ($Prog$) is low, the capacitor is connected to the 1T1R Delay device, whose Bit Line is set at $Vref$, forming an $RC$ pair. During the operation of the circuit, the Delay 1T1R Word Line is open (set at 4.8 V). For this reason, the capacitor's voltage $Vcap$ recharges with a time constant $\tau = RC$. A Thresholding unit detects when $Vcap$ relaxes back to the resting potential, crossing a certain threshold voltage, in our case set at 250 mV. Note that the Thresholding unit is bypassed when the IN spike is applied so that only the second crossing of the threshold by the $Vcap$ potential is detected. More details on the Thresholding unit are in Supplementary Information, Fig. 3a. The output of the Thresholding unit is a spike of the same shape as the input one, just at 4.8 V. This voltage pulse passes through the second Multiplexer set by the selection $Prog$ voltage, entering the second section of the Dendritic Circuit. The output spike ($V_{OUT}$), which occurs with a certain delay compared to the input spike IN, is applied to the Word Line of the Weight 1T1R. The Bit Line of the Weight RRAM is pinned at the RRAM's reading voltage (around 0.6 V).

As a consequence, an output current $I_{OUT}$ is generated, proportional to the conductance of the Weight RRAM. This current is read out and then fed to an output LIF neuron (details in the Supplementary Information, Fig. 3b).

**Fabrication/integration.** We fabricated our circuits in a 130 nm technology, in a 200 mm production line. The RRAM devices' stack is $TiN/Si:HfO/Ti/TiN$, formed by a 10 nm thick $Si:HfO$ layer sandwiched by two 4 nm thick $TiN$ electrodes. Notably, we selected a thicker oxide layer so that the pristine state's resistance would be maximized while the Si doping reduces the forming voltage. Also, local Si implantation has been demonstrated to increase the resilience of RRAM devices at high temperatures[79]. We perform a retention test in Supplementary Fig. S2 in Supplementary Note 1, in Supplementary Information.

Each RRAM device is coupled to an access transistor, forming 1T1R structures, that are used to select RRAM devices individually during programming operations. The size of the access transistor is 650 nm wide. RRAMs are built between metal layers 4 and 5, allowing to integrate them in the Back-End-of-Line, maximizing integration density.

To access RRAMs in DenRAM, peripheral circuits have been designed. Each array row and column (Word Line, Bit Line, and Source

Line) is interfaced by a multiplexer connecting the line either to ground or to a pad, where a programming/reading voltage could be applied. Multiplexers are operated by Shift Registers that store the addresses of the lines to be connected to the pad's programming/reading voltage. All the circuits are featured in a 200 mm wafer and are accessed by a probe card connecting to pads of size of [50 × 90] µm² each.

### RRAM characteristics

Smart programming of the devices can be reached to obtain more precise conductance levels and stabilize the devices with respect to the filament relaxation resulting in a conductance shift[57,59].

The resistive switching mechanisms employed in our paper's devices are based on the creation and dissolution of a conductive pathway within the device, brought about by the application of an electric field. This change in the pathway's geometry leads to distinct resistive states within the device. To execute a SET or RESET operation, we apply either a positive or negative pulse across the device, respectively. This pulse formation or disruption within the memory cell results in a decrease or increase in its resistance. When the conductive pathway is formed, the cell is in the Low Resistive State (LRS); otherwise, it is in the High Resistive State (HRS). In a SET operation, the bottom of the 1T1R structure is conventionally held at ground level, while a positive voltage is applied to the top electrode of the 1T1R. Conversely, in a RESET operation, the reverse is applied.

### Dendritic circuit measurement setups

The tests of the circuit involved analyzing and recording the dynamical behavior of analog CMOS circuits as well as programming and reading RRAM devices. Both phases required dedicated instrumentation, all simultaneously connected to the probe card. For reading the RRAM devices, Source Measure Units (SMU)s from a Keithley 4200 SCS machine were used while, for programming, a B1530A waveform generator by Keysight was used in order to send SET and RESET pulses. To maximize the stability and precision of the programming operation, SET and RESET are performed in a quasi-static manner. This means that a slow rising and falling voltage input is applied to either the Top (SET) or Bottom (RESET) electrode, while the gate is kept at a fixed value.

To the $V_{top}(t)$, $V_{bot}(t)$ voltages, we applied a trapezoidal pulse with rising and falling times of 50 ns, a pulse width of 1 µs and picked a value for $V_{gate}$. For a SET operation, the bottom of the 1T1R structure is conventionally left at ground level, while in the RESET case the $V_{top}$ is equal to 0 V and a positive voltage is applied to $V_{bot}$. Typical values for the SET operation are $V_{gate}$ in [1.6–2.2]$V$, while the $V_{top}$ peak voltage is normally at [2–2.6]$V$. Such values allow modulating the RRAM resistance in an interval of [8–50]kΩ corresponding to the LRS of the device. For the RESET operation, the gate voltage is instead at 4.5 V, while the bottom electrode is reaching a peak at [1.8–2.4] V.

The HRS is less controllable than the LRS due to the inherent stochasticity related to the rupture of the conductive filament, thus the HRS level is spread out in a wider [60–1000]kΩ interval. The reading operation is performed by limiting the $V_{top}$ voltage to 0.4 V, a value that avoids read disturbances, while opening the gate voltage at 4.5 V.

Inputs and outputs of the dendritic circuit and the LIF neuron are analog dynamical signals. In the case of the input, we have alternated a HP 8110 pulse generator with a B1530A by Keysight combined with a B1500A Semiconductor Device Parameter Analyzer by Keysight. As a general rule, input pulses had a pulse width of 1 µs and rise/fall time of 20 ns. This type of pulse is assumed as the stereotypical spiking event of a Spiking Neural Network. Concerning the outputs, a 1 GHz Teledyne LeCroy oscilloscope was utilized to record the output signals of the various elements in the dendritic circuit and the LIF neuron. An Arduino Mega 2560 board for collecting statistics on read-to-read delay variability was used when testing the dendritic circuit alone,

using its built-in timers for recording the delay between the input spike and the output spike, varying the timer scale in order to adapt to the delay order of magnitude and avoid overflow problems in the board registers. Due to the impossibility of reading resistances as high as pristine (in the order of tens to hundreds of GΩ), the measures of the pristine resistances are extracted from the delay measurements on 71 dies through the formula $R_i = D_i/C$.

### RRAM-aware noise-resilient training

The strategy of choice for endowing DenRAM with the ability to solve real-world tasks is hardware-aware gradient descent. The conventional backpropagation-based optimization utilized in machine learning is tailored to the hardware substrate implementing DenRAM. In particular, DenRAM makes use of RRAMs as synaptic weights: this imposes constraints on the parameters of the network, that are accounted for during the training phase. The main challenge is the stochasticity of RRAMs, resulting in overprecise weights.

We propose to address this problem by introducing the non-idealities of RRAM during the training phase. During inference, we perturb the weights of the neural network with the same variability measured on the RRAM devices. However, we apply the computed gradients to the original unperturbed weights. This methodology is similar to Quantization-Aware-Training[80]. Eventually, we perform a pre-training phase without RRAM's variability, so as to better initialize the hardware-aware training. We also expand such a training procedure in the case of faulty Weight-RRAM devices. We found that DenRAM re-trained in such a way is resilient to broken RRAM devices, as shown in Supplementary Fig. S8 in Supplementary Note 4.

### Heartbeat anomaly detection task description

For the Heartbeat anomaly detection task, we chose the MIT-BIH dataset[52]. Such a database is composed of continuous Electro-Cardio-Gram (ECG) 30-min recordings measured from multiple subjects. Each ECG recording is annotated by different cardiologists indicating each heartbeat as normal or abnormal, and the type of arrhythmia, when present. The data for each subject contains the recording from two leads, but it has been demonstrated that one lead is sufficient for performing a correct classification[81].

For the ECG task, patient 208 has been selected having the most balanced label count between normal and abnormal heartbeats of all the subjects. Labels have been grouped in normal heartbeats (labels "L", "R" and "N") and anomalies (labels "e", "j", "A", "a", "J", "S", "V", "E", "F", "/", "f" and "Q"). The 30-min recording has been divided into segments of 180 time steps each around the R peak[82] and divided into a train set and a test set of equal sizes. The input channel coming from the measurements of lead A has been converted into spikes through a sigma-delta modulator[61] generating two different inputs for the NN: one with the so-called up spikes and the other with the down spikes.

This data is fed to either the DenRAM architecture or an SRNN, and outputs are encoded as the spiking activity of the single output neuron. Large output spiking activity - above a predefined threshold - signals the detection of arrhythmia.

### Keyword spotting task description

The SHD dataset[25] is based on Lauscher artificial cochlea model that converts audio speech data to spike train representation with 700 channels, similar to spectrogram representation with Mel-spaced filter banks. It consists of 10,000 recordings (8156 training, 2264 test samples) for 20 classes of spoken digits from zero to nine in both English and German languages. We divided the original training dataset into according to a 80–20% train-val split. Each recording duration in the dataset is maximum 1.4 s and converted spikes are time binned into 280 5 ms bins, resulting in (700 × 280) dimensionality. We observed only 2% of the samples have spikes after the 150th timestep, thus we truncated the input duration to 750 ms. Only on the simulations with

256 dendritic arbors, we sampled three times along the channel dimension without overlap to obtain three augmentations with 256 channels. This sampling improved the speed of our simulations by reducing the total number of parameters in the network and increased the both sizes of training and testing datasets. The network is trained using BPTT with the cross-entropy loss where the logits are calculated using the maximum potential over time non-linearity for each leaky-integrator output neuron. We use a batch size of 64 for delay-based architectures and 128 for SRNN. Each experiment reported was conducted using 3 distinct random seeds. All experiment hyperparameters (membrane decay time constants, spike threshold, weight scaling and learning rates) are tuned separately to obtain maximum performance.

## Data availability
The MIT-BIH ECG dataset[52] and the Spiking Heidelberg Datasets[25] dataset are publicly accessible. All other measured data are freely available upon request.

## Code availability
All software programs used in the Article are available at https://github.com/EIS-Hub/DenRAM.git.

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

## Acknowledgements
We acknowledge funding support from the H2020 MeM-Scales project (871371) (F.M., G.I., E.V., M.P.), Swiss National Science Foundation Starting Grant Project UNITE (TMSGI2-211461) (M.P.), Marie Skłodowska-Curie grant agreement No 861153 (Y.D., G.I.), as well as European Research Council consolidator grant DIVERSE (101043854) (E.V.). We are grateful to Jimmy Weber for helpful discussions throughout the project.

## Author contributions
M.P. proposed the idea. M.P., F.M., E.V., and G.I. developed the dendritic circuit and network concepts. F.M. and M.P. designed and laid out the circuits for fabrication. E.V. and L.G. supervised the fabrication of the circuits in hybrid RRAM-CMOS process. S.D. and N.C. performed the characterizations and verification on the fabricated circuits. S.D. developed the hardware-aware training of the dendritic network for heartbeat anomaly benchmark. T.T. developed the hardware-aware training of the dendritic network for the keyword spotting task. Y.D. implemented the first delay implementation on JAX and supervised the hardware-aware simulations. All authors contributed to the writing of the manuscript. M.P. and E.V. supervised the project.

## Competing interests
The authors declare no competing interests.
