## [Peer Review File · Nature Communications]

REVIEWER COMMENTS

Reviewer #1 (Remarks to the Author):

This manuscript proposes a new type of neuromorphic architecture inspired by the dendritic organization in the biological brain. In my opinion, this work is of great relevance, particularly for the proposed concept inspired by neuroscience that supports well temporal processing of dynamic inputs as well as the coincidence detection experimental demonstration in RRAM/130nm CMOS. I believe this manuscript is well written and the results in Fig. 4 show significant promise. The manuscript is worthy of publication after revisions and would benefit the community.

Some requests for clarification from the authors and suggestions for revisions are included below:

- Biological neurons seem to have quite a variety of dendritic trees depending on connectivity. In the proposed architecture, is it possible to have dendritic branches of different lengths for the same input or a different number of dendritic branches for input?
- Are the hardware-aware simulation results for limited bit precision RRAM weights?
- What would be the impact of stuck on RRAM weights on the DenRAM performance? What about pristine delay RRAM devices that have outlier low resistance values?
- In the proposed architecture, only the weight RRAM are trainable, with the delay RRAM providing random values around the measured mean. However, the biological synaptic delays seem to be capable of modulation. What are the advantages / disadvantages for the proposed architecture of having random vs. trainable delay elements? It would be helpful to talk more on the discussion section on the potential expansion and applicability of the proposed architecture.
- On line 161, it is mentioned that when the second input arrives too early or too late compared to the signal in one of the dendritic branches of the first input, no coincidence is detected. Is there a quantitative metric of what constitutes the threshold for coincidence?

Reviewer #1 (Remarks on code availability):

The code does not seem available at this time, only a readme and a license file. It says that "This repo will be updated soon."

Reviewer #2 (Remarks to the Author):

The work by D'Agostino et al. introduces a novel RRAM-based architecture for neuromorphic computing, with a focus on dendritic delays in coincidence detection. The work is novel, highly relevant and promises significant value for the neuromorphic community. I endorse its publication in Nature Communications but have a few suggestions:

1. I could not find the value of the Vcap capacitor. What is the limit of a practical capacitor value that would yield longer delays, even with lower resistance states of the RRAM element?
2. Abstract: Minor point - RRAM stands for Resistive RAM, which is distinct from the broader term 'resistive memory.'
3. Could the authors briefly elaborate on what specific aspects of dendritic functionality are missing from studies 24-28? Are dendrites completely omitted in these studies?
4. Fig 2c may require a minor adjustment, as it depicts three voltages (Vi, Cap, and Vout), yet only two axes are shown for Vcap and Vout. The authors might consider adding additional label.
5. In Caption 2f, it might be clearer not to refer to multiple resistive states as LRS and HRS, since this implies only two states, rather than multiple.
6. Are the differences between 'delay RRAM' devices and 'Weight RRAM' devices solely in their resistance states (pristine, LRS/HRS), or are there other distinctions? I presume that only 'delay RRAM' benefits from very high resistance, unlike 'weight RRAM.', so is there any benefit of using different RRAM technologies for these?
7. Regarding RRAM devices, why does Si doping lead to reduced electroforming? Could the specific electroforming voltage be provided (noting >3V in Si; is it much higher than 3V)? What are the pulse widths used for programming? While high endurance may not be critical in this context, what is the typical endurance for devices used? Are Si-doped devices inferior to pure HfOx-RRAM devices in any specific parameter?
8. For 'delay RRAM' devices, are they ever programmed post-fabrication, or are they solely used as fabricated? Given the necessity for high pristine resistance, is HRS "high enough"? Is there any disadvantage to having a high spread of pristine resistances that cannot be more accurately programmed, or if the devices are electroformed, the pristine state cannot be achieved anymore?
9. Are there specific retention tests for 'weight RRAM' devices, and do these tests relate to the devices studied in <https://doi.org/10.1002/aisy.202200145>? Specifically, are the devices in the prior study Si-doped, and does Si-doping affect retention or relaxation? Have you conducted any analysis to determine if the conductance, especially in pristine and HRS states, varies with temperature? Electronic conduction might be enhanced by temperature. Could this be an issue?
10. In the outlook section, the note that RRAM devices exhibit 'Ohmic conduction' might not be fully accurate; particularly for HRS and pristine states, where conduction is highly non-linear and not Ohmic.

This paper is excellent, and with the clarification of these points, it should be ready for publication.

Response to the reviewers:

DenRAM: Neuromorphic Dendritic Architecture with RRAM for Efficient Temporal Processing with Delays

Dear reviewers,

We sincerely thank you for your insightful comments on our manuscript. We believe addressing the comments has made the paper much stronger and more clear, and we are grateful for that. Below, we reply to each concern of your concerns one by one. We have addressed the changes in the main and supplementary texts, highlighted with a strike-through red when text was removed, and in blue when text was added. In this letter, we reply with black, and point out the action we took for each of your comments. Whenever the changed text is copied here, we highlight it in green for your attention. Thank you very much.

Response to Reviewer 1

This manuscript proposes a new type of neuromorphic architecture inspired by the dendritic organization in the biological brain. In my opinion, this work is of great relevance, particularly for the proposed concept inspired by neuroscience that supports well temporal processing of dynamic inputs as well as the coincidence detection experimental demonstration in RRAM/130nm CMOS. I believe this manuscript is well written and the results in Fig. 4 show significant promise. The manuscript is worthy of publication after revisions and would benefit the community.

Reply 1 Thank you very much for acknowledging the relevance of our work and its significant promise. Below, we do our best to address each of your questions and concerns.

Some requests for clarification from the authors and suggestions for revisions are included below: - Biological neurons seem to have quite a variety of dendritic trees depending on connectivity. In the proposed architecture, is it possible to have dendritic branches of different lengths for the same input or a different number of dendritic branches for input?

Reply 2 [Response] Thank you for raising this point concerning biological plausibility. The simple answer is yes, we can control the size of dendritic branches in DenRAM. We can achieve this both in the design phase and by means of the training algorithm.

In our model we consider the length of a dendrites can be interpreted in two different ways.

- In a biological context, the delay associated with a dendritic branch depends on multiple factors, among which is the length of dendrites. This is because the farther in space is the input, the more distance the spike has to travel and thus the longer the delay. By having randomly distributed delays we implicitly emulate heterogeneous dendrites length.
- In a software implementation context, the length of the dendrites can be considered as the number of synapses per dendrite (which in our simulations is fixed). To implement length heterogeneity we can prune a given proportion of synapses according to the absolute value of the associated weights. This yields a network where the input channels are delayed a random number of times (instead of a fixed common number of times). Figure R1a, shows the accuracy of DenRAM on the heartbeat anomaly detection task as a function of the proportion of removed synapses. In Fig. R1b, we show that we can prune up to 40% of the synapses, corresponding to the lowest absolute weights, and get tolerable accuracy drops of around 2% for both architecture sizes (D1 and D2) utilized in the keyword spotting task. Figure R1c shows the resulting distribution of the number of synapses per dendritic branch with 10% (blue) and 20% (yellow) of the synapses pruned. The resulting distribution shows heterogeneity in the number of synapses per branch, similar to the biological counterparts. Moreover, we find that when pruning is applied, indeed inputs can end up having different number of branches. For example, if 40% of the synapses, corresponding to the lowest absolute values, are pruned out, some delay branches completely disappear, which reduces the number of branches for some inputs.

[Action] In the main text, lines [295-299], we have added an explanation, that by acting on the Weight-RRAM we can disable dendritic connections in DenRAM, potentially implementing biologically inspired connectivity patterns. We have also added the quantitative analysis on pruning the DenRAM architecture (Figure R1) in Supplementary Note 4.

- Are the hardware-aware simulation results for limited bit precision RRAM weights?

Reply 3 [Response]: Thank you for allowing us to clarify this point about the hardware-aware simulations. We follow a methodology first developed in the NeuRRAM paper published last year on Nature¹, which similar to DenRAM, tackles the

Figure R1. Effect of heterogeneity in dendritic length. a) Removing the synapses with the smaller associated absolute weight for the ECG task. b) Removing the synapses with the smaller associated absolute weight. We sweep the percentage of synapses removed. c) Number of synapses per branch after removal for architecture D2, for different removal proportions.

problem of imprecise weights in a neural network, implemented by RRAM devices. The solution is an RRAM-aware training procedure whereby the noise produced during the RRAM programming operation is introduced in the learning loop. This procedure assumes that the weights in the network are affected by Gaussian noise with a given standard deviation, due to the imprecise programming operation in RRAMs. Experimental results confirm that this assumption fits the conductance distributions when programmed with progressively stronger SET operations, as is done in Fig. 2f of the main text. A single SET operation is applied to all the devices in a 16 kbit RRAM array, while progressively increasing the programming compliance current. Overall, the standard deviation in the distributions is about 10% of the total conductance range (minimum conductance of $1\mu\text{S}$, maximum conductance of $65\mu\text{S}$). Note that by using a program-and-verify scheme, the width of these distributions can be lowered to less than 5% of the conductance range².

Therefore, in our simulations, we allow the RRAMs to assume any value in the $[1-65]\mu\text{S}$ range, with 10% noise. The conductance value of RRAM is sampled from a Gaussian distribution with standard deviation determined by the experimental array-level characterization (see Fig. 2f, main text) being at $6.5\mu\text{S}$.

During training, we inject such RRAM-calibrated noisy distributions in the weights for the forward pass, while evaluating the network accuracy, and utilize the original non-noisy weights for the weight updates. Please see³ for the benefit of this training method. Once the offline training phase is complete, we freeze the trained weights. For the testing phase, we apply Gaussian noise calibrated on the RRAM-programming results to the trained weights and evaluate the performance on the testing set of the task at hand, repeating such a procedure 5 times. We report the mean and standard deviation of the accuracy on the testing set.

[Action] We reformulated the explanation of the Hardware-Aware training in line 175. In particular, we avoided the term “programming resolution” which could be misunderstood as quantized weight values in RRAMs. Instead, we mention that

programming RRAM devices results in the distribution of conductances, with certain mean and standard deviation, that are mapped and utilized during training, so to make the network more robust to the non-idealities of RRAM devices during inference.

- What would be the impact of stuck on RRAM weights on the DenRAM performance? What about pristine delay RRAM devices that have outlier low resistance values?

Reply 4

[Response] Dear Reviewer, thank you for this question. To assess the reliability of DenRAM we originally characterized its resilience to noise in the weight. Your question allows us to widen such an analysis. RRAM failure could occur by becoming stuck either in a High Resistive State (HRS) or a Low Resistive State (LRS). In the following analysis, we assume the two failure types to have equal probability. We perform RRAM simulation on the networks trained for Fig. 4 of the main text, artificially introducing failures in the Weight RRAMs. First, we gauge the effect of faulty RRAMs on the classification accuracy, assuming that the model is trained without any prior knowledge of the RRAM failure. The results are reported in Fig. R2 under PTS (Post Training Stuck). We find that the larger network trained on the SHD task (Fig. R2b) is more resilient to PTS, compared to the smaller network that was used for the ECG task (Fig. R2a). We then examine the effect of faulty RRAMs on the classification accuracy, when the network is re-trained, aware of the faulty devices. The results are reported in Fig. R2 under SAT (Stuck Aware Training). We find that the accuracy on the ECG task is almost fully restored to the original case without RRAM failures while accounting for up to 25% faulty devices. Similarly, DenRAM applied to the SHD task shows little accuracy degradation for up to 30% of faulty RRAM devices. These results confirm that DenRAM is not only energy-efficient and parameter-efficient, but is also resilient to faulty RRAM devices, especially if the network is aware of the device yield. Regarding the possible outliers in the Delay-RRAM resistance distribution, we agree that this is an important aspect to account for, given that the Pristine State in RRAM cannot be controlled. We analyzed the role of the Delay-RRAM resistance distribution in Figure S5 and S6 in Supplementary Note 4. First, we fit the Delay-RRAM distribution as a log-normal (Figure S5a) distribution and changed the mean of such distribution, looking at the effect on classification accuracy. We then repeated such procedure for a distribution with either a smaller or larger standard deviation (Figure S5b, Figure S6a,b). Note that increasing the standard deviation of the distribution involves including more outliers compared to the original distribution. In the results, we note that this is not in general problematic, rather we notice that the heterogeneity of Delay RRAM can be beneficial, especially if the mean delay value is low. Also, DenRAM allows us to "avoid" outlier delay values not used for the task at hand by selecting their associated weights to zero (High-Resistive-State in the Weight RRAM). For these reasons, we show that DenRAM is robust to outlier resistance values in the Delay-RRAM.

Figure R2. Effect of faulty weights in the ECG task and SHD tasks. PTS: post-training stuck, where the network does not know about device failure *a priori*; SAT: stuck-aware training, where the network is aware of the failed devices. a) Analysis of network resilience on the ECG task. b) Analysis of network resilience on the SHD task and for two different network sizes (D1 and D2).

[Action] We added the results presented in Figure R2 in Supplementary Note 4, in the Supplementary Information. We refer to this Supplementary Note at Lines [428-429] of the main text.

- In the proposed architecture, only the weight RRAM are trainable, with the delay RRAM providing random values around the measured mean. However, the biological synaptic delays seem to be capable of modulation. What are the advantages / disadvantages for the proposed architecture of having random vs. trainable delay elements? It would be helpful to talk more on the discussion section on the potential expansion and applicability of the proposed architecture.

Reply 5

[Response] Thank you for this interesting question. The DenRAM architecture was conceived around Resistive Random Access Memory (RRAM) devices implementing both weights and delay parameters. As the High-Resistive State (HRS) of RRAMs is poorly controllable, and its average resistance is too low to implement long delays, we developed DenRAM to feature multiple Dendritic Circuits in each branch, exploiting the distribution of very large resistance of RRAMs in the pristine state (as demonstrated in Supplementary Figure 4). Despite the poor control on the resistance of RRAM devices in their HRS, DenRAM is capable of leveraging the large variety of delays generated by the multiple Dendritic Circuits, reaching competitive results in both the ECG and SHD tasks. We discovered that the “duplication effect” of the dendritic branches is a useful asset for a shallow network, like DenRAM.

However, we believe that a device that can be precisely controlled at the $G\Omega$ range could be very beneficial for DenRAM. If the delay elements are precisely controllable, we can leverage the training of the delay elements, thus finding the optimal delay values. We envision that by focusing on optimal delay values and avoiding sub-optimal delays - currently present in the pristine state RRAM distribution - we can reduce the size of the dendritic branches, optimizing the area footprint and power consumption of DenRAM.

Future versions of DenRAM may benefit from featuring devices such as Ferroelectric Tunnel Junction (FTJ) devices⁴, which promise controllable, $G\Omega$ -level resistance values. We are excited by the prospect of improving the performance of DenRAM with novel devices!

[Action] We added a paragraph in the Discussion section (lines [321-325]) to explain how DenRAM would benefit from exploiting novel devices to implement the Delay element and training the delays. The text is copied below for your convenience.

Training delays Utilizing the FTJ technology in DenRAM would result in enhanced delays and better control over the delay values in the dendritic branches. This opens to the possibility of learning the delays with gradient descent, as performed in⁵. Optimizing the delay values, it is foreseeable that the number of required Dendritic Circuits per dendritic branch would reduce drastically, further lowering the parameters of DenRAM. As a consequence, we envision that DenRAM could further improve its performance and power efficiency by exploiting novel devices, enabling the training of delays.

- On line 161, it is mentioned that when the second input arrives too early or too late compared to the signal in one of the dendritic branches of the first input, no coincidence is detected. Is there a quantitative metric of what constitutes the threshold for coincidence?

Reply 6 [Response] Dear reviewer, we are happy to clarify this point. As the Coincidence Detection (CD) mechanism in the neuron's dendrites finds the temporal correlations between the different inputs, the neuron is only activated, when two or more of its inputs arrive in close temporal proximity. If the spikes arrive close to each other in time, such that the membrane potential of the neuron passes the threshold, the neuron is activated. Therefore, the membrane time constant of the neuron defines the temporal window in which the input spike proximity is defined. We define the largest temporal difference between input spikes that activate the neuron as the CD window. If the membrane voltage decays at a fast rate, the CD window shrinks, otherwise it widens. In the original experiment in Fig. 3d, we set the time constant of neurons at $\tau = 10ms$, showing that this produces a CD window of about 6ms. We repeated the experiments with decaying time constants of $\tau = \{5, 15, 20\}ms$ and, as expected, the CD window widens to approximately $\{3, 10, 15\}ms$ (Fig. R3). The exact value of the CD window is determined by the time constant, yet noise and variability in the analog neuromorphic circuit, and the noise in the RRAM weights generate some variability.

To summarize, we highlight that CD is performed by crossing the threshold voltage of the output neuron. The DenRAM architecture allows to control the center of the CD window thanks to the delays in the Dendritic Branch. In Fig. 3b we show that such a CD window is centered around 58ms, the value of the D_1 delay. This unique delay functionality of DenRAM is leveraged in the processing of complex signals, such as in the case of the Electrocardiogram and Speech (results of Fig. 4).

[Action] We repeated the experiment in Fig. 3d of the main text, while varying the time constant of the post-synaptic neuron, and report the result in the Fig R3. We included this figure in the Supplementary Note 2, as well as an explanation of the meaning of the CD window.

Figure R3. Coincidence Detector implemented with DenRAM. With the same setup described in Fig. 3 of the main text. New measurements were added to the Coincidence Detector experiment, sweeping the time constant of the output neuron. We show that the time constant of the post-synaptic neuron's membrane potential modulates the CD window. This temporal window is the temporal distance between the two input spikes for which the Coincidence Detector neuron activates its output spike.

The code does not seem available at this time, only a readme and a license file. It says that "This repo will be updated soon."

Reply 7

[Response] Dear Reviewer, we took the time to review our code and make sure it runs smoothly and provides the results that feature in the paper. We are satisfied with the result and hope you appreciate the effort we put into making our code available, readable, and written in a nice form. Note that we include a "requirement.txt" file with all the packages to install to run our experiments. The actual code is found in the "code" folder. You can experiment with the "main.py" python file which trains DenRAM on the SHD task, by default with a network of 700 inputs and 16 delays per branch (D1 configuration as in Figure 4 main text). You can find information on how we perform the training loop in "training.py", while the structure of the network is defined in "utils_training.py". We also include a plotter file ("plot_fig4d_D1andD2").

[Action] We added our code and the "requirement.txt" file in the GitHub repository in the following link:
https://github.com/EIS-Hub/DenRAM/blob/master/code/utils_training.py

Response to Reviewer 2

The work by D'Agostino et al. introduces a novel RRAM-based architecture for neuromorphic computing, with a focus on dendritic delays in coincidence detection. The work is novel, highly relevant and promises significant value for the neuromorphic community. I endorse its publication in Nature Communications but have a few suggestions:

Reply 8 Thank you very much for the acknowledgement of the relevance of our paper. We are very glad that you liked it!

1. I could not find the value of the Vcap capacitor. What is the limit of a practical capacitor value that would yield longer delays, even with lower resistance states of the RRAM element?

Reply 9

[Response] Dear Reviewer, this question points to an interesting technological consideration of great importance for the DenRAM architecture. We settled for a capacitor value of around 400 fF, which was a trade-off between the required area and the achieved delays. Coupled with the Pristine Resistive State of RRAMs reaching hundreds of GΩ, our RC circuit yields tens of milli-second time constant and thus delays.

The capacitor value is generally a function of the technology node, and the capacitance density of the available capacitors in the technology. We opted for a 130 nm CMOS node with RRAM devices in the Back-end of Line (BEOL) as an available technological platform to demonstrate DenRAM. In this technology, high-density Metal Insulator Metal (MIM) or Metal

Oxide Metal (MOM) capacitors are not available, since the RRAM devices are integrated between the fourth and fifth metal layers. Therefore, we have used the capacitor from the gate oxide of the transistors (MOS Cap), whose capacitance density is $3.7 \text{ fF}/\mu\text{m}^2$ ⁶.

Therefore, the practical limitation is the size of the synaptic cell, which is dominated by the capacitor size. Currently, our design uses $10 \times 10 \mu\text{m}^2$ for 400 fF of MOS caps.

Improving the underlying technological platform of DenRAM can massively improve the potential of the architecture. First, MIM/MOM capacitors would allow to shrink the size of the Dendritic Circuit embedding the capacitor on the BEOL. Furthermore, the time constant of the dendritic RC can be increased if the resistance of the RC circuit can be increased, allowing us to decrease the capacitance. Currently, the resistance value is limited by the RRAM devices, even though we have used them in their Pristine State. The prospect of featuring different memory technologies with higher resistance, such as FTJ devices ⁶, is extremely intriguing as it would enable a large (and controllable) dendritic delay while reducing the size of the capacitor.

Moreover, having lower capacitor values, would in turn have power benefits. As producing dendritic delays involves the charging of the Capacitor, the energy spent charging the capacitor scales as:

$$E_{cap} = \frac{1}{2} CV^2 \quad (1)$$

where C is the capacitance, and V is the voltage of the capacitor. As a consequence, we foresee that increasing the dendritic resistance R is a better strategy to yield maximal delays while minimizing the area footprint and the power consumption.

[Action] We have specified the type of Capacitor utilized in the Dendritic Circuit, as well as its capacitance value in lines [82-83] in the main text. We copy it here for your convenience.

Concerning the capacitor in the Dendritic Circuit, we utilize a conventional transistor gate oxide capacitance, setting the capacitance C to around 400 fF.

2. Abstract: Minor point - RRAM stands for Resistive RAM, which is distinct from the broader term 'resistive memory.'

Reply 10 [Response] We thank the Reviewer for the clarification. We made sure to refer to RRAMs as "resistive random access memories", especially when referring to the devices in DenRAM.

[Action] We modified the "resistive memory" terminology to "resistive RAM" in lines [44, 166, 337, 346].

3. Could the authors briefly elaborate on what specific aspects of dendritic functionality are missing from studies 24-28? Are dendrites completely omitted in these studies?

Reply 11

[Response] Thank you for this question, which gives us the opportunity to say with details how our work is going beyond the state of the art. Below we are providing a summary of each work and say how that is different from DenRAM.

[24] presents an analog Very Large Scale Integration (VLSI) implementation of passive and active dendrites with dendritic spikes. Delays are briefly mentioned in the context of CD, where the highly synchronized spikes travel much faster through the dendritic line than other spikes, thanks to dendritic spikes. However, delays are not used as a useful feature for computation, which is the key feature of the DenRAM architecture.

[25] presents an analog VLSI implementation of a local learning rule based on the post-synaptic activity, where the authors use a constant delay between pre-synaptic and post-synaptic spikes. Unlike DenRAM, this does not feature delays as a useful means for computation.

[26] presents an analog computing array with neuron blocks implemented using Floating Gate (FG) technology. Each neuron is paired with a dendritic array that can emulate dendritic spikes and cable attenuation, which will delay the signal through a chain of FG-based elements. The FG devices could not be programmed to very small currents and thus the circuit performs 10 times faster than was required for processing speech signals (similar to DenRAM). However, FG technology has limitations in terms of slow speed of programming and large required programming voltages. DenRAM exploits the features of novel memory technologies to overcome these limitations, and can achieve time constants up to 60 ms which is required for real-time sensory processing.

[27] presents an implementation of passive and active dendrites with dendritic potentials on BrainScaleS-2. The delays are implemented using spatial arrangement of the synapses, and are on the range of micro-seconds. Therefore, unlike DenRAM, these delay values will not support real-time sensory processing applications.

[28] presents the only memristor-based implementation of dendrites. However, although the dendrites feature non-linearity, they do not feature delays, which is the highlight of the DenRAM architecture.

[Action] We changed the sentence in lines [36-37] to better highlight the limitations of previous work. We copy it here for your convenience.

However, most of these implementations either do not use delays as a variable for computation, or use technologies that have do not support real-time processing due to their fast time scales.

4. Fig 2c may require a minor adjustment, as it depicts three voltages (V_i , V_{cap} , and V_{out}), yet only two axes are shown for V_{cap} and V_{out} . The authors might consider adding additional label.

Reply 12

[Response] Thank you for pointing out the lack of clarity in Fig. 2c. We intentionally color-coded the curves plotting Delay Capacitor voltage (V_{cap}) and the output spike pulse (V_{OUT}), and plotted them on the right and left y-axes with matching color codes, as the two traces span different scales (0.6V and 5V respectively). The Input voltage is applied at time=0, with a magnitude of 1.2 V, which is a value that lies between the scale of the two y-axis. However, we agree that such a voltage trace is hardly visible. Therefore, to clarify this, we removed the IN trace from Fig. 2c and explained in its caption how the input voltage pulse is applied to the circuit at time $t=0s$.

[Action] We removed the input pulse gray trace (IN) from Fig. 2c. Instead, we added a clarification sentence about the application of the IN pulse in the caption of Fig. 2c.

5. In Caption 2f, it might be clearer not to refer to multiple resistive states as LRS and HRS, since this implies only two states, rather than multiple.

Reply 13

[Response] Thank you for clarifying this point about the definition of the different resistive levels. To avoid ambiguity, following your suggestion, we now simply refer to “different resistive levels” when describing Fig. 2f. The same nomenclature is kept throughout the rest of the paper.

[Action] We changed the caption of Fig. 2f and removed the reference to “different LRS levels” throughout the paper line 104.

6. Are the differences between 'delay RRAM' devices and 'Weight RRAM' devices solely in their resistance states (pristine, LRS/HRS), or are there other distinctions? I presume that only 'delay RRAM' benefits from very high resistance, unlike 'weight RRAM.', so is there any benefit of using different RRAM technologies for these?

Reply 14

[Response] Dear Reviewer, we confirm that the Delay- and Weight-RRAM in DenRAM currently employ the same type of device, RRAM devices. As you correctly pointed out, the difference is in the way that the Delay- and Weight-RRAMs are programmed. For the Delay-RRAM, we aim at maximizing the delay, thus requiring large resistance. This is achieved by leaving the Delay-RRAM in the pristine state. For the Weight-RRAM, we have a different set of requirements. Ideally, the Weight-device has a large On/Off ratio, precise programming operation (low cycle-to-cycle variability), large retention of the resistive state, and low conductance relaxation after programming. We find RRAM devices to comply well enough with such requirements, both as Delays and as Weights, as demonstrated in the hardware-aware simulation results in the main text (Section 2.2).

Potentially, there are devices with different features than RRAMs that might represent a better fit for the role of either Delay or Weight devices. For example, FTJ technology^{4,6} assures large controllable resistance, thus being appealing as Delay elements. FeFET technology^{7,8} promises better control over the resistance level compared to RRAMs, representing a good prospect as a Weight device. However, integrating different technologies in the same wafer represents an additional and non-negligible cost. Having said that, such heterogeneous integration in the BEOL is a promising and interesting direction which is recently gaining traction in the Non-Volatile-Memory community⁹. A very interesting prospect would be to integrate the access transistor of the Delay and Weight devices in the BEOL, to enable even greater integration density.

To summarize, DenRAM could benefit from employing a diverse set of devices for the Delay- and Weight-RAMs. While we already discussed in the main text and in reply 5 the possible benefits of FTJ as Delay device, we emphasize that increasing the precision of Weight devices can be beneficial in improving performance (Figure 4). Despite the non-negligible additional cost associated with heterogeneous integration, we envision that DenRAM can benefit from a diverse set of devices and further improve its computational power and power efficiency.

[Action] We added a paragraph in Outlook (lines [326-332]) to explain the requirements of the Weight devices and how different technologies, such as FeFET, could improve the performance of DenRAM. For your convenience, we are copying the paragraph below.

Weight precision As highlighted in Fig. 4d, the precision of the weight is an important parameter in determining the performance of DenRAM. RRAM devices offer relatively good control over the conductance levels by modulating the Word Line (WL) voltage during the SET programming operation. Yet, such programming operations in RRAMs yield a distribution of conductance with about 10% of standard deviation, relative to the maximum conductance (Fig. 2f). To improve the precision of weights in DenRAM, advanced read-verify programming operations on RRAMs can be performed^{1,10}. Alternatively, different memory technologies, with better control over their conductance levels, can be used. An example technology is FeFET, demonstrating up to 5-bit precision in its conductance levels^{7,8}.

7. Regarding RRAM devices, why does Si doping lead to reduced electroforming? Could the specific electroforming voltage be provided (noting >3V in SI; is it much higher than 3V)? What are the pulse widths used for programming? While high endurance may not be critical in this context, what is the typical endurance for devices used? Are Si-doped devices inferior to pure HfOx-RRAM devices in any specific parameter?

Reply 15

[Response] Dear Reviewer, thank you for the question to clarify the properties of the used RRAM devices, central to the working of the DenRAM implementation in this paper. Regarding the choice of Silicon implantation, earlier works provide a thorough analysis of Si-doped HfO₂ RRAM devices, and the decrease of the forming voltage is attributed to the appearance of Si interstitial defects^{11,12}. Figure 7 of¹¹ (copied below as Fig. R4 left) highlights that Si implantation both of Local Implantation (LI) (blue symbols) and Blanket Implantation (BI) (red symbols) lower the Forming Voltage. In Table IV in the same paper (copied below as Fig. R4 right), it is reported that Si implantation do not cause any decrement in Endurance.

	REF.	LI (5% Si)	BI (5% Si)
R_{PRISTINE} (Ohm)	>1E7	2E6	2.5E4
V_F (V) (100 ns pulses)	4	2.1	2.7
V_{SET/RESET} (Q.S.) (V)	0.8 / 1.1	0.7 / 0.9	1.05 / 1.4
MW_{MEDIAN}	100	100	4
MW_{3σ}	<1	10	<1
RETENTION BER 1000 min @165°C	LRS: 1E-3 HRS: 2E-3	LRS: <1E-4 HRS: 7E-4	LRS: 8E-2 HRS: 5E-2
Endurance	>10 ⁷	>10 ⁷	<10 ⁷

Figure R4. Left) Median forming voltage evolution and dispersion of OxRAM devices in a 4 kb array with Si fraction for BI and LI devices in pulsed mode (100 ns pulse, voltage ramp). Right) Comparison between no-implant reference, BI, and LI. Both images are reproduced from¹¹.

In our programming setting, the Forming operation requires voltage pulse timing of 50 ns-10 ms-50 ns for rise-width-fall times, and the voltage is higher than 3 V (with a medium value around 4 V). We adopted a sequential programming procedure where multiple programming pulses are applied, increasing the programming voltage each time, until the RRAM device is formed. SET and RESET operations have much shorter programming pulses: 50 ns-1 μs-50 ns for rise-width-fall.

[Action] We specified the programming voltage pulses for Forming operation in Supplementary Note 1, in the Supplementary Information.

8. For 'delay RRAM' devices, are they ever programmed post-fabrication, or are they solely used as fabricated? Given the necessity for high pristine resistance, is HRS "high enough"? Is there any disadvantage to having a high spread of pristine resistances that cannot be more accurately programmed, or if the devices are electroformed, the pristine state cannot be achieved anymore?

Reply 16

[Response] Dear Reviewer, indeed, the role of the resistance spread of the Delay RRAM is a characteristic of DenRAM that interested us while conceiving the architecture. In most experiments we performed, all the Delay RRAM devices are left in the pristine state which corresponds to the fabrication condition obtained before the first forming operation. Once the devices are formed, they cannot return to the Pristine State.

Through simulations, we found that for both the tasks analyzed in this paper, i.e. heartbeat arrhythmia detection (ECG) and spoken digit classification (SHD), the High Resistive Level was not sufficient to achieve high classification accuracy. HRS has a resistance of about $100k\Omega$, which coupled with the capacitor of $400fF$, implemented in DenRAM, produces delays that are not long enough for the analyzed tasks. This of course does not exclude the possibility that there might be other tasks which can be solved by shorter delays, for which the HRS resistance is a good match.

Therefore, we used the devices in their pristine state, thoroughly analyzing the effect of the resistance distribution on the accuracy of DenRAM in the given tasks. We originally reported these results in the Supplementary Information.

We start by fitting a log normal distribution to the statistical measurements of RRAM's pristine state (Figure R5a, standard deviation=0.5, green curve). We then vary the standard deviation to study the effect of different resistance spread on the accuracy of the network for the tasks at hand (plotted in Figure R5a in purple and blue for $\sigma=0.25$ and 0.75 respectively). We analyze the classification accuracy on the ECG and SHD tasks using these two distributions, while also varying the mean delay. The results are shown in Fig. R5b for the ECG task, and in Fig. R5c,d for the SHD task. We see that larger variability in the delays is beneficial and improves performances when the mean delay is low. We find that the variability of the Pristine State of RRAM not only is not a limiting factor, but is rather a feature leveraged by the DenRAM architecture.

[Action] We added a line in Supplementary Note 1 to clarify that once the devices are formed, they cannot go back to the Pristine State. We also added a clarification on Supplementary Note 4 on the requirement of resistance for the Delay RRAM when tackling the ECG and SHD tasks. This is then a segue to the presentation of the ablation study on the effect of the Delay-RRAM distribution from Figure R5.

9. Are there specific retention tests for 'weight RRAM' devices, and do these tests relate to the devices studied in <https://doi.org/10.1002/aisy.202200145>? Specifically, are the devices in the prior study Si-doped, and does Si-doping affect retention or relaxation? Have you conducted any analysis to determine if the conductance, especially in pristine and HRS states, varies with temperature? Electronic conduction might be enhanced by temperature. Could this be an issue?

Reply 17

[Response] We acknowledge the concerns regarding retention and relaxation in RRAM devices, as they are crucial to the proper functioning of DenRAM. Figure R6 presents data retention measurements at room temperature for our RRAM devices, starting from various initial conductance levels. It is observed that the highest conductance levels exhibit a drift toward lower conductance values, amounting to a decrease of 10% in the worst case. To mitigate the impact on the network performance caused by this drift effect, we have incorporated it into the RRAM-aware training procedure we conducted. The results depicted in Fig. 4 of the main text demonstrate that DenRAM exhibits robustness to conductance deviations in weights up to 10% of the RRAM conductance range.

The relaxation affects both HfO_2 (as in²) and Si-doped RRAM devices. To mitigate the short-term relaxation effect, we have implemented a technique based on program-and-verify loops with wait times of 1 s, as proposed in².

We agree with the reviewer, that electronic conduction may be influenced by temperature. However, previous analysis on HfO_2 RRAM devices demonstrated that the High Resistive State remains stable up to 200° (see Figure 5 in¹³, green plot corresponds to TiN/Ti contacts equivalent to those in our RRAM devices).

[Action] We have added the reference¹¹ in the Method section, pointing out that Local Implantation of Si has been demonstrated to increase the stability of the High-Resistive-State in RRAMs. Lines [372-373]. We also added the Retention measurement in Supplementary Note 1, in Supplementary Information.

Figure R5. Ablation study on DenRAM evaluating the spreading of the pristine resistance distribution. Weights are subject to 10% noise. a) Lognormal distribution of the Delay RRAM (R_d). We denote σ as the standard deviation of the underlying normal distribution of RRAMs. The RRAM data are fit with $\sigma = 0.5$. b) Accuracy on the MIT-BIH dataset as a function of the mean of the Delay RRAM's distribution, for different distributions. c) Accuracy as a function of the mean of the Delay RRAM's distribution, using 256 inputs and 8 delays using the SHD dataset. d) Accuracy on the SHD dataset as a function of the mean of the Delay RRAM's distribution, for different distributions, using 700 inputs and 16 delays.

10. In the outlook section, the note that RRAM devices exhibit 'Ohmic conduction' might not be fully accurate; particularly for HRS and pristine states, where conduction is highly non-linear and not Ohmic.

Reply 18

[Response] Thank you for the clarification. We agree that the conduction mechanism in HRS and the pristine state in RRAMs is not ohmic. Therefore, we removed "Ohmic Conduction" in the phrase in the Outlook section.

[Action] We removed "Ohmic conduction" from lines [319-320].

Figure R6. Retention measurements of the multi-level programming conditions. We plot the mean of the distributions obtained from a 16kb RRAM array, programmed at different average conductances. Measurement of the conductance level of the RRAM array is repeated over time, up to 24 h.

This paper is excellent, and with the clarification of these points, it should be ready for publication.

Reply 19 Thank you very much. We are thrilled to hear you find our work excellent.

References

1. Wan, W. *et al.* A compute-in-memory chip based on resistive random-access memory. *Nature* **608**, 504–512 (2022).
2. Esmanhotto, E. *et al.* Experimental demonstration of multilevel resistive random access memory programming for up to two months stable neural networks inference accuracy. *Adv. Intell. Syst.* **4**, 2200145, DOI: [10.1002/aisy.202200145](https://doi.org/10.1002/aisy.202200145) (2022).
3. Moro, F. Memristor-aware-training for resilient neural networks (2023).
4. Covi, E. *et al.* Ferroelectric tunneling junctions for edge computing. In *2021 IEEE International Symposium on Circuits and Systems (ISCAS)*, 1–5, DOI: [10.1109/ISCAS51556.2021.9401800](https://doi.org/10.1109/ISCAS51556.2021.9401800) (2021).
5. Shrestha, S. B. & Orchard, G. SLAYER: Spike layer error reassignment in time. In Bengio, S. *et al.* (eds.) *Advances in Neural Information Processing Systems 31*, 1419–1428 (Curran Associates, Inc., 2018).
6. Pillonnet, G. & Jeannot, N. Effect of cmos technology scaling on fully-integrated power supply efficiency. In *CIPS 2016; 9th International Conference on Integrated Power Electronics Systems*, 1–5 (VDE, 2016).
7. Grenouillet, L. *et al.* Reliability assessment of hafnia-based ferroelectric devices and arrays for memory and ai applications (invited). In *2023 IEEE International Reliability Physics Symposium (IRPS)*, 1–8, DOI: [10.1109/IRPS48203.2023.10118099](https://doi.org/10.1109/IRPS48203.2023.10118099) (2023).
8. Müller, F. *et al.* Multilevel operation of ferroelectric fet memory arrays considering current percolation paths impacting switching behavior. *IEEE Electron Device Lett.* (2023).
9. Ramaswamy, N. *et al.* Nvdram: A 32gb dual layer 3d stacked non-volatile ferroelectric memory with near-dram performance for demanding ai workloads. In *2023 International Electron Devices Meeting (IEDM)*, 1–4, DOI: [10.1109/IEDM45741.2023.10413848](https://doi.org/10.1109/IEDM45741.2023.10413848) (2023).
10. Esmanhotto, E. *et al.* High-density 3d monolithically integrated multiple 1t1r multi-level-cell for neural networks. In *2020 IEEE International Electron Devices Meeting (IEDM)*, 36.5.1–36.5.4, DOI: [10.1109/IEDM13553.2020.9372019](https://doi.org/10.1109/IEDM13553.2020.9372019) (2020).
11. Barlas, M. *et al.* Improvement of hfo2 based rram array performances by local si implantation. In *2017 IEEE International Electron Devices Meeting (IEDM)*, 14–6 (IEEE, 2017).
12. Barlas, M. *et al.* Impact of si/al implantation on the forming voltage and pre-forming conduction modes in hfo2 based oxram cells. In *2016 46th European Solid-State Device Research Conference (ESSDERC)* (IEEE, 2016).
13. Traoré, B. *et al.* Hfo 2-based rram: Electrode effects, ti/hfo 2 interface, charge injection, and oxygen (o) defects diffusion through experiment and ab initio calculations. *IEEE Transactions on Electron Devices* **63**, 360–368 (2015).

REVIEWERS' COMMENTS

Reviewer #1 (Remarks to the Author):

The paper has been reviewed satisfactorily. In particular, it is good to see that the proposed DenRAM is flexible in the configuration of the dendritic branches and robust to outlier resistance values in the Delay-RRAM. The retention plot and the code also valuable additions. I don't have any further suggestions for the paper.

Reviewer #1 (Remarks on code availability):

The code is well organized and documentation is provided regarding its installation and usage.

Reviewer #2 (Remarks to the Author):

The authors have duly answered my questions, and I have no reservations about recommending the manuscript for publication in its current form.

This paper will be a good addition for the community working on memristor-based neuromorphic systems.